# Perceived Racial Discrimination, Psychological Distress, and Suicidal Behavior in Adolescence: Secondary Analysis of Cross-Sectional Data from a Statewide Youth Survey

**DOI:** 10.3390/healthcare12101011

**Published:** 2024-05-14

**Authors:** Meredith Cahill, Robert Illback, Nicholas Peiper

**Affiliations:** 1REACH Evaluation, Louisville, KY 40208, USA; 2Department of Epidemiology and Population Health, University of Louisville, Louisville, KY 40202, USA; nicholas.peiper@louisville.edu

**Keywords:** racial discrimination, psychological distress, suicidal ideation, suicide attempts, adolescence

## Abstract

Developmental, clinical, and epidemiological research have demonstrated the salience of perceived racial discrimination (PRD) as a contributor to negative mental health outcomes in adolescence. This article summarizes secondary analyses of cross-sectional data from a large-scale youth survey within a predominantly rural state, to estimate the prevalence and strength of the association between PRD and serious psychological distress (SPD), suicidal ideation, and prior suicidal attempts. Data from 93,812 students enrolled in 6th, 8th, 10th, or 12th grade within 129 school districts across Kentucky were examined, to determine prevalence rates for subgroups within the cohort. Logistic regression analyses assessed the differences and established comparative strength of the association among these variables for racial/ethnic subgroups. PRD was self-reported at high rates across several demographic subgroups and was most evident among Black (24.5%) and Asian (22.1%) students. Multiracial students experienced the highest rates of both SPD and suicidality (ideation and prior attempt). Both for the entire cohort and for each racial/ethnic subgroup, PRD was significantly associated with an increased likelihood of negative mental health outcomes, although the strength of these associations varied across the subgroups and developmental levels. The implications for early intervention and prevention are discussed.

## 1. Introduction

### 1.1. Developmental Context

Adolescence is a critical timeframe for social–cognitive and neurodevelopmental processes that set the stage for later growth, development, and adjustment [1]. It is a time of vulnerability and change, during which physical and emotional transformations occur, identities and social perspectives are formed, relationships are shaped, and worldviews are built [2]. Given that cumulative risk theories of development support the potency and long-lasting impact of adolescent experiences [3], an appreciation for how racial discrimination can shape the adolescent experience for racial and ethnic minorities is crucial. This is especially salient at a time when youth mental health, in general, is of heightened societal concern, given the disruptions associated with the global pandemic. Recent data from the CDC’s 2021 Adolescent Behaviors and Experiences Survey of a representative nationwide sample of adolescents in grades 9–12, for example, showed that 44.2% experienced persistent feelings of sadness and hopelessness in the prior year, 19.9% had seriously considered attempting suicide, and 9.0% reported having attempted suicide [4].

Perceived racial discrimination (PRD) in adolescence has been recognized as a contributor to negative mental health outcomes [5,6,7,8,9,10]. As a construct, PRD denotes a subjective, experiential attribution that occurs when an individual perceives they have been treated unfairly or unfavorably based on their race/ethnicity. Although race and ethnicity are two separate social constructs, race refers to the concept of dividing people into groups based on various sets of physical characteristics (e.g., skin color) and the antiquated process of ascribing biological differences between those groups, often for political purposes, to maintain white hegemony [11,12], while ethnicity describes the culture of people in a given geographic region, including their language, heritage, religion, and customs [13]. In societies with diverse populations and a long history of racial/ethnic subjugation such as in the U.S., these distinctions become blurred when individuals may identify with or be perceived to identify with multiple racial or ethnic groups [14]. Consequently, the two constructs, though conceptually distinct, are often linked in terms of experiences of PRD, as research indicates that racism rather than race is the most relevant risk factor for poor health outcomes [15,16,17].

PRD is associated with experiences such as the expression of negative, harmful, and inaccurate stereotypes or beliefs; avoidable or unfair inequalities of power, resources, or opportunities; emotions (fear and hatred), behaviors, or practices (unfair treatment and threats) that have a negative impact; and related interpersonal or systemic phenomena (e.g., political disempowerment, segregation, and racial wealth gap) that influence development, health, opportunities, and life course [18,19].

A recent comprehensive meta-analytic review noted that perceptions of difference and discrimination are discernible even in early life experience [20]. Infants and young children recognize differences in race and ethnicity and can perceive in-group preferences at preschool age [21]; by middle childhood, cultural stereotypes related to skin color and other-group prejudices are identifiable [22]; overt discriminatory actions are recognized by children as young as 10 [23]; in early adolescence, young people begin to recognize and label their own racial and ethnic identity and can discern how they are viewed by others [24]; and by late adolescence, as abstract thinking emerges, concepts such as racism, privilege, and the impacts of discrimination come into awareness. At this stage of growth, young people are especially vulnerable to the effects of overt discrimination [25].

The experience of PRD is embedded in a broader social–ecological framework, within which individual personal identities and perceptions intersect with forces such as gender, social class, and sexuality to shape health and well-being outcomes [26,27]. Minority Stress Theory (MST) has been widely employed as an explanatory model to link stress induced by PRD with negative mental health outcomes in marginalized populations [8,28,29]. This form of stress is distinct from everyday stressors experienced during the developmental period and accrues from a mixture of persistent stressful experiences that may be proximal (e.g., situational expectations of discrimination, negative remarks, or stereotypes) or distal (e.g., experience of general prejudices or discriminatory practices). Persistent exposure to stressors associated with PRD have been found to diminish psychological resilience, alter patterns of behavior, and weaken emotional control, leading to a greater vulnerability for poor mental health [10].

### 1.2. Clinical Research

Research on the linkages between PRD and mental health in the United States has primarily focused on adult population subgroups, including Native Americans [30], Black or African Americans [31], Asian Americans [32], and Hispanic Americans [33]. Over the past decade, however, extensive clinical evidence has emerged to document the mechanisms and impact of PRD, as it shapes the health and well-being of adolescents. For both adults and adolescents, PRD has been associated with disparities in health, achievement, identity formation, and adjustment across the lifespan [18,19,34,35].

Within the meta-analysis referenced above, peer-reviewed research was examined in 214 studies encompassing 489 unique effect sizes for 91,338 adolescents [20]. A clear and consistent pattern was found, in which PRD among minority adolescent subpopulations was associated with higher levels of depression and internalizing symptoms (e.g., anxiety, loneliness, and stress), psychological distress, lower achievement and engagement, externalizing behaviors, risky behavior, substance abuse, and association with deviant peers. More specifically, in addition to a range of mild-to-moderate problems in living, associated with stress induced by PRD (e.g., anxiety or social problems), clinical research has established consistent and clear relationships between PRD and more serious problems, including psychological distress and suicide-related behavior [20]. A review of 138 empirical population-based studies of self-reported racism and various health/mental health outcomes, after adjusting for confounding variables (e.g., age, education, income, and health risks), found that the strongest and most consistent evidence emerged for associations between PRD and psychological distress and depressive symptoms [29].

A prospective study on the longitudinal trajectories of a sample of 504 urban African American adolescents in Grades 7–10 used growth-mixture modeling to explore the stability and change in PRD over time. The investigators found three trajectories of discrimination—increasing, decreasing, and stable low. Boys were more likely to be in the increasing group and to experience racial discrimination as they aged. Adolescents in the increasing group were four times more likely to experience depressive symptoms and twice as likely to exhibit aggressive behaviors. As PRD increased for adolescents within this trajectory, their level of psychological distress and clinical sequelae increased concomitantly [36].

There is also evidence of a growing disparity in mental health outcomes, as demonstrated by recent population-level trends in suicide-related behavior among adolescents. During the past decade, the reported prevalence of suicidal ideation and planning, suicide attempts, and death by suicide has begun to decline within the general adolescent population [37]. In contrast, rates for racial/ethnic minority adolescents for these indicators have increased during this timeframe, heightening the concern for the vulnerability of adolescent minority subgroups [38,39]. In this context, some researchers have suggested that the available data for racial/ethnic minority adolescents may underestimate the prevalence of suicide-related behavior within these communities [40,41,42].

### 1.3. Epidemiological Research

International population-based research consistently confirms the impact of PRD on distress and suicidality among racial and ethnic adolescent subgroups. An assessment of longitudinal UK data from the Determinants of Adolescent Social Well-Being and Health survey found that for a large sample of young people in London schools, racism was associated with diminished well-being trajectories for all ethnic groups [43]. In a national cross-sectional survey in New Zealand, associations were demonstrated between perceptions of racial and ethnic discrimination experienced by indigenous and minority students and the experience of negative health and well-being outcomes such as depressive symptoms, substance misuse, feeling safe, and achievement [44]. Among African Canadian adolescents in British Columbia, annual surveys conducted between 2003 and 2018 found that PRD had gradually increased over that period; girls and immigrant African Canadians were most likely to have experienced discrimination, while adverse mental health outcomes associated with PRD varied across population subgroups (immigrant African Canadians and Black males were most likely to experience negative and persistent mental health problems) [45].

Similarly, within the US, secondary analysis of data from the Adolescent Brain Cognitive Development study deployed multivariate regression models to test the association of PRD with suicidality. Researchers sought to determine the prevalence of self-reported PRD, disentangling the unique contribution of PRD to suicidality, independent of other variables and circumstances (e.g., adversities, environmental exposures, demographics, and/or other biological and developmental conditions). They also sought to assess the variability of impact among various racial and ethnic subgroups. The results of this study showed that Black American young people experienced higher levels of PRD and more suicidality than other racial groups, independent of other possible contributing factors. Although the strength of the association was not as strong as the findings for Black adolescents, an association between PRD and suicidality was found for other racial subgroups (but the dataset lacked sufficient statistical power to make clear determinations for Asian, Native American, and Native Hawaiian participants). The authors concluded that PRD imposed distress associated with suicidality at a level similar to other well-established risk factors [46].

Secondary analyses were conducted on data obtained from the CDC’s 2021 Adolescent Behaviors and Experiences Survey, collected during the COVID pandemic, on a stratified nationally representative sample of students in grades 9–12 in the US (*n* = 7705). Lifetime PRD was found to be most prevalent among Asian, Black, and multiracial students (63.9%, 55.2%, and 54.5%, respectively) and, to a lesser extent, Native Hawaiian/Other Pacific Islander (48.5%) and Hispanic/Latino (41.5%). These rates of PRD contrasted with White/Non-Hispanic (22.5%) and American Indian/Alaska Native (26.7%) students. PRD was associated with poor negative mental health outcomes and low levels of social connectedness to a far greater extent, especially for Asian and Black students [47].

In a separate secondary analysis of data from this survey, data from a sample of 3241 minority adolescents were analyzed using logistic regression. After controlling for other factors (e.g., self-identification as lesbian/gay, bi-sexual, or questioning; cyberbullying; feeling sad or hopeless; and poor mental health during the pandemic—all well-established risk factors for suicidality), racial/ethnic adolescents who reported PRD evidenced 1.57 times higher odds of experiencing suicidal ideation, 1.64 times higher odds of making a suicide plan, and 1.67 times higher odds of attempting suicide during the past year [5].

The current study presents secondary analyses of cross-sectional data from a large-scale survey of young people in 6th, 8th, 10th, and 12th grade from a predominantly rural state (Kentucky). It further explores epidemiological estimates of prevalence rates and multivariate associations between PRD with psychological distress and suicidality across racial/ethnic groups and developmental stages. By examining a sample that diverges from typical national samples, the study aimed to expand upon the current literature by providing a more in-depth understanding of PRD and mental health issues from a place-based perspective. Through this approach, the study not only sought to examine potential variations in findings from our sample compared to other large-scale studies, but also to provide further evidence for targeted public health and prevention interventions addressing PRD among adolescents, particularly in Kentucky and states with similar demographic compositions.

## 2. Materials and Methods

### 2.1. Study Population

The current study examines cross-sectional, self-reported data from the Kentucky Incentives for Prevention Youth Survey (KIP), a statewide behavioral health survey of 6th-, 8th-, 10th-, and 12th-grade students in participating Kentucky School Districts. Administered for over two decades by the Division of Substance Use Disorder in the Cabinet for Health and Family Services, the survey aims to anonymously assess student use of alcohol, tobacco, and other drugs, as well as additional factors related to social and emotional well-being, including school safety, mental health, and bullying. The core survey items were initially curated by the US Substance Abuse and Mental Health Services Administration (SAMHSA), informed by extensive research on the risk and protective factors associated with substance abuse among children and adolescents [48]. This framework allows for inter-state and national comparisons, as well as intra-state regional analyses.

Historically, the survey has been administered biennially during the fall of even-numbered years (e.g., 2014, 2016, and 2018). However, owing to disruptions caused by the COVID-19 pandemic, the most recent administration of KIP was postponed from fall 2020 to fall 2021. Beginning with the 2021 iteration, the survey transitioned to a classroom-based administration, utilizing a web-based platform.

Participation in the survey, both at the school district and individual student levels, has always been voluntary, incurring no cost to the involved school districts. Rigorous measures were implemented to uphold student anonymity and mitigate any potential coercion to participate. The survey utilized a passive consent procedure, whereby parents were informed of the survey administration, purpose, and content, allowing them to decline participation on behalf of their child.

In 2021, KIP was conducted within 129 of Kentucky’s 171 school districts, resulting in a 75% participation rate. In the 129 districts that participated, 93,812 6th-, 8th-, 10th-, and 12th-grade students completed the survey. Appendix A displays the demographic features of the 2021 survey respondents in comparison to enrollment data from the Kentucky Department of Education (KDE) for the 2020–2021 school year. In 2021, Kentucky’s largest, most urban, and most diverse school district, Jefferson County Public Schools, did not participate in KIP. School district enrollment data including and excluding Jefferson County data are, therefore, provided for comparison. Overall, while grade and gender comparisons are similar, there is a notable variation in racial and ethnic composition, reflecting the more urban environment of Jefferson County. The most pronounced difference noted when comparing total enrollment with and without Jefferson County is for the proportion of non-Hispanic Black students (11.3% vs. 6.7%, respectively).

### 2.2. Measures

#### 2.2.1. Serious Psychological Distress

The K6 scale includes six items designed to gauge the frequency of experiencing internalizing symptoms related to psychological distress over the preceding 30 days. These items assess self-perceived emotional states of feeing (1) “nervous”, (2) “hopeless”, (3) “restless or fidgety”, (4) “so depressed that nothing could cheer you up”, (5) “that everything was an effort”, and (6) “worthless” [49]. Respondents rated their experiences on a scale ranging from “never” to “all of the time”. Responses were coded numerically as 0 to 4, yielding an unweighted summary scale ranging from 0 to 24 [50]. A K6 score of ≥13 was considered indicative of experiencing serious psychological distress in the past 30 days, in accordance with established guidelines [50]. The K6 has been extensively validated in general population samples of adolescents, including on several US federal surveillance surveys [51]. On the KIP and a similar youth survey, the K6 has demonstrated adequate psychometric properties for generating prevalence estimates to inform the design and evaluation of public health interventions [52,53].

#### 2.2.2. Suicidal Behaviors

Suicidality over the past 12 months was measured by assessing both suicidal ideation and suicide attempts. Suicidal ideation was evaluated using a yes-or-no question, asking students if they had seriously considered attempting suicide in the past 12 months. Suicide attempts were measured by asking students how many times they had attempted suicide in the past 12 months. Responses pertaining to suicide attempts were dichotomized into the following two categories: zero attempts or one or more attempts during the 12-month time frame. These items were adapted from a US federal surveillance survey (Youth Risk Behavior Survey) and have adequate psychometric properties [54,55,56,57].

#### 2.2.3. Perceived Racial Discrimination

In 2021, students were asked whether they had been a target of racism in the past 12 months, to which they could respond with Yes, No, or Not Sure. This question was adapted from a youth survey in the US, conducted by the National 4-H Council in partnership with The Harris Poll, and was subsequently modified based on feedback derived from youth focus groups conducted within the state [58]. Students who answered ‘Yes’ to this question were categorized as having experienced racial discrimination in the past year. Response options were dichotomized for the analysis, with ‘No’ and ‘Not Sure’ responses categorized together.

#### 2.2.4. Demographic Factors

Several self-reported sociodemographic factors were included in the analyses, including race/ethnicity, sexual orientation, and gender identity (SOGI). Racial/ethnic categories included Non-Hispanic (NH) White, NH Black, Hispanic, NH Asian, NH Native Hawaiian or Pacific Islander, NH Alaska Native or American Indian (AN/AI), NH Other, and NH Multiracial. Due to the small number of NH Native Hawaiian or Pacific Islander respondents (*n* = 148), these responses were grouped with NH Asian students in the analyses. For ease of reading, these racial/ethnic groups will be referred to, moving forward, as White, Black, Hispanic Asian/Pacific Islander, Native American, Other, and Multiracial, with the recognition that race is a social construct and that these groups and labels do not comprehensively represent the diverse identities of student respondents.

Gender identity options included male, female, identity not listed here, questioning or unsure, and prefer not to say, which was expanded from the binary categorization used in previous administrations. Included for the first time in 2021, sexual orientation was categorized as straight/heterosexual, gay/lesbian, questioning or unsure, other identity not listed here, and prefer not to say. Grade (6, 8, 10, or 12) and school level (middle school or high school) were also incorporated into the analyses. Rurality was determined based on the 2021 Federal Office of Rural Health Policy’s rural county designations [59]. Responses were categorized as rural if the student attended school in a county eligible for rural health funding as of 1 October 2021.

### 2.3. Data Analysis

Descriptive statistics were calculated for demographic characteristics, PRD, and mental health outcomes. Cross-tabulations were then conducted to examine the prevalence of PRD and the three mental health outcomes, in relation to race and ethnicity. Logistic regression models stratified by race/ethnicity were fit to examine the multivariate association between PRD with psychological distress, suicide ideation, and suicide attempt, with grade level, gender identity, sexual orientation, and rurality entered as covariates. Firth’s penalized likelihood method was used to ameliorate issues related to model separation and biased parameter estimates due to small cell sizes [60]. This yielded adjusted odds ratios (aORs) and 95% confidence intervals for PRD and the demographic characteristics associated with the mental health outcomes for each racial and ethnic group. As a sensitivity analysis, we replicated all the models separately for middle and high school levels. A two-sided *p*-value of 0.05 was used as a threshold for statistical significance, although we used the 95% CIs to make inferences about effect sizes and precision, rather than solely relying on *p*-values that are more sensitive to reaching significance in large samples. Stata Version 17 was used to conduct the analyses.

## 3. Results

### 3.1. Study Population

Table 1 provides a summary of the characteristics of the 93,812 6th-, 8th-, 10th-, and 12th-grade students from Kentucky who participated in the 2021 KIP Youth Survey. Among these respondents, slightly over half were in middle school (55.7%), with 49.1% identifying as male, 44.8% as female, and 5.7% as outside the gender binary (questioning/unsure, identity not listed, or prefer not to say). The sample was made up of predominantly White (71.2%), followed by Hispanic (8.8%) and Black (6.3%) students. In terms of sexual orientation, the majority identified as heterosexual (67.0%), with 5.8% identifying as gay/lesbian and 7.0% as an unlisted identity. Over half of the students (55.5%) attended school in rural counties. Appendix A provide the characteristics of middle and high school students, respectively.

### 3.2. Prevalence of Perceived Racial Discrimination

Among the entire sample of students, 5.9% reported PRD in the past year. Figure 1 shows the prevalence of PRD by race/ethnicity. Approximately one-fourth of Black students reported PRD (24.5%), followed closely by Asians/Pacific Islanders at 22.1% and Multiracial students at 19.0%. White students reported the lowest rate of PRD at 2.8%. Appendix A illustrates the prevalence of PRD by both race/ethnicity and school level.

### 3.3. Prevalence of Serious Psychological Distress and Suicidality

Table 1 summarizes the overall prevalence of serious psychological distress (SPD), suicide ideation, and suicide attempts, as well as by demographic categories. One in five (20.6%) students reported SPD, while 13.4% experienced suicidal ideation and 7.4% reported one or more suicide attempts. Additionally, larger proportions of sexual- and gender-minority students, students attending schools in rural counties, and students experiencing PRD reported SPD and suicidal behavior. Figure 2 shows the mental health outcomes by race and ethnicity. For SPD, the highest rate was observed for Multiracial students (27%), followed by Native Americans (21.8%), Other (21.3%), and Hispanics (21.2%). For suicidal ideation, Multiracial students again emerged with the highest overall rate (19.2%), while slightly lower rates were observed amongst the other groups (range 11.6–14.3%). For suicide attempts, most racial/ethnic subgroups of students reported similar rates in the 8.5–11.5% range, but White and Asian/Pacific Islander students evidenced lower rates (6.6% and 5.9%, respectively). School-level variations are depicted in Appendix A.

### 3.4. Multivariate Associations between PRD and Mental Health

Figure 3 illustrates the results from the multivariable associations between PRD and each of the mental health outcomes, both overall and stratified by race and ethnicity. Overall, PRD was significantly associated with SPD (aOR: 2.37, 95% CI: 2.22–2.53), suicidal ideation (aOR: 2.48, 95% CI: 2.31–2.65), and suicide attempts (aOR: 2.70, 95% CI: 2.51–2.95), after adjusting for social and demographic characteristics. Moreover, the association remained significant across all racial and ethnic groups. (Note that in all instances, 1.0 is not present within the confidence interval, lending further confidence to the finding, per the criterion described for data interpretation in the Methods section). 

Notable differences were observed between subgroups. The subgroups with the highest strength of association between PRD and SPD were Native American (aOR = 3.64, 95% CI: 1.96–6.74) and Asian (aOR = 3.26, 95% CI: 2.34–4.55). A similar pattern was observed for the association between PRD and suicidal ideation, with the largest effect sizes again amongst Native Americans (aOR = 4.77, 95% CI: 2.58–8.82) and Asians (aOR = 4.82, 95% CI: 3.25–7.13). For PRD and suicide attempts, White students (aOR: 3.20, 95% CI: 2.77–3.69) evidenced the strongest association and highest likelihood, followed by Asians/Pacific Islanders (aOR = 2.93, 95% CI: 1.72–4.99), Blacks (aOR = 2.71, 95% CI: 2.16–3.39), and Native Americans (aOR = 2.71, 95% CI: 1.34–5.50). 

Sensitivity analyses revealed variations in the strength of the association between PRD and mental health outcomes by school level. With the exception of the association between PRD and suicide attempts among Asians/Pacific Islanders, larger effect sizes were observed among middle school students than high school students, for all three outcomes. The association between PRD and SPD (Appendix A) was significantly higher among white middle school students (aOR = 3.82, 95% CI: 3.22–4.52) than white high school students (aOR = 2.52, 95% CI: 2.18–2.90). For suicidal ideation (Appendix A), the association with PRD was significantly higher among Black middle school students (aOR = 3.93, 95% CI: 3.06–5.03) compared to Black high school students (aOR = 1.78, 95% CI: 1.35–2.35). For suicide attempts (Appendix A), the association with PRD was significantly higher for middle school students overall (aOR = 3.27, 95% CI: 2.92–3.65) compared to high school students overall (aOR = 2.24, 95% CI: 1.99–2.52), but no differences were observed by school level, among the racial and ethnic groups. 

## 4. Discussion

### 4.1. Key Findings

The present study explored the relationships between perceived racial discrimination (PRD) and three negative mental health outcomes—past 30-day serious psychological distress (SPD), past-year suicidal ideation, and past-year suicide attempts in a large, primarily rural sample of adolescents (N = 93,812). The study aimed to investigate potential variations in findings compared to other large-scale studies. Additionally, it aimed to provide further evidence for targeted public health and prevention interventions addressing PRD among adolescents, particularly in Kentucky and states with similar demographic compositions. Overall, 5.9% of the sample indicated that they perceived racial discrimination within the past year. For the mental health outcomes, 20.6% reported experiencing SPD in the past month, 13.4% experienced suicidal ideation in the past year, and 7.4% reported attempting suicide during the same period. In the stratified analyses, multiracial students reported the highest overall rates of SPD and suicidality; Asian and Black students reported the lowest rates of SPD; Asian students evidenced the lowest rates of suicidal ideation; and Asian and White students reported the fewest suicidal attempts. These disparities support growing national evidence of heightened vulnerability among adolescent minority subgroups, particularly multiracial individuals [38].

Similar to findings from other national surveillance programs, the prevalence of PRD varied by race/ethnicity among this group of Kentucky students. Black, Asian, and multiracial students reported the highest PRD rates compared to other groups, aligning with recent secondary analyses of CDC’s 2021 Adolescent Behaviors and Experience Survey data. However, the lifetime prevalence rates of PRD among the CDC sample were notably higher than those reported over the past year from high schoolers in our study (35.6% versus 7.3%, respectively), possibly due to differences in the time period for which PRD was measured [47]. Consistent with the existing epidemiological and clinical literature, PRD was significantly associated with an increased likelihood of serious psychological distress, suicidal ideation, and suicide attempt across the entire sample and each racial/ethnic group [5,18,19,20,35,36,43,44,45,46,47]. As with the aforementioned 2021 CDC study, these results remained consistent even after controlling for other known demographic risk factors for suicide such as SOGI and rurality, though the strength of association between PRD and suicidality in the current study was larger in magnitude than the results of the CDC analysis [5].

Though PRD was significantly associated with SPD and suicidality across all racial/ethnic groups, there were notable subgroup differences. Specifically, Native American and Asian/Pacific Islander students exhibited larger effect sizes for both SPD and suicidal ideation, despite Native American students reporting relatively low levels of PRD and Asian/Pacific Islander students having a relatively low prevalence of mental health issues. Additionally, multiracial students showed high rates of PRD and mental health issues, but comparatively lower associations. The reasons for these subgroup differences are not fully understood, as few studies have been able to provide definitive findings regarding PRD and its association with serious psychological distress and suicidality among indigenous, Asian/Pacific Islander, and multiracial youths. Similar to our findings, the 2007 study of secondary school survey data from New Zealand, found that indigenous, Asian, and Pacific Islander students who experienced racial discrimination were more likely to report depressive symptoms [44]. The meta-analysis conducted by Benner et al. (2018) found that PRD has a larger effect size on Asian youth’s social–emotional well-being, in comparison to other subgroups [20] Multiracial students also reported high levels of PRD in the 2021 CDC study, but differences in effect sizes between different racial/ethnic groups were not assessed [5,47].

It is possible that the subgroup differences are partly attributable to both contextual and personal characteristics that have been shown to moderate the experience of PRD. For example, an inquiry into the impact of school and neighborhood diversity on perceptions of racial discrimination identified varied and complex patterns of influence on self-esteem and life satisfaction [61]. Similarly, an investigation of negative effects associated with PRD in African American adolescents found that they varied in relation to the number of contexts within which discrimination was experienced. The severity of impact was also moderated using available familial support [62]. Racial centrality (the extent to which an individual considers race to be a central component of their identity) was also found to influence the level of distress experienced, both in terms of heightening the experience of discrimination, but also as a buffer of negative impact [63,64]. While the 2021 KIP Youth Survey did not capture these types of personal and contextual characteristics, future research should examine their potential moderating effects on the relationship between PRD and mental health among adolescents.

Another key finding from the current study was that school level moderated the associations between PRD and all three mental health outcomes. Stronger associations between PRD and SPD and suicidal behavior were observed among middle schoolers across the entire sample and for most racial and ethnic groups. Similar results were found by Benner et al. (2018), who reported that the association between PRD and socioemotional distress was stronger in early adolescents (10–13), and by Astell-Burt et al. (2012), who found that the effect size between PRD and psychological well-being decreased with age [20,43]. It is possible that the moderating effect of school level may be due to coping strategies developed through experience and increasing maturity [65]. A review of coping literature documents developmentally graded differences in effective coping skills across childhood and adolescence, with early adolescence being a period of far less skills-based resiliency than the later adolescent years [66]. It may also be the case that by mid-adolescence, there has been more time for youths to develop strategies for coping with PRD that buffers the associated mental health outcomes.

### 4.2. Implications for Prevention and Intervention

The results of the current study provide clear evidence of the need for culturally tailored interventions addressing PRD and mental health within schools and communities, particularly for interventions beginning in or before early adolescence. An influential integrative conceptual framework for targeted prevention and intervention with minority young people marginalized by PRD was initially advanced by García Coll and colleagues [67]. Subsequent developments in developmental science broadened the framework to expand from narrow risk reduction approaches (e.g., coping strategies) toward multi-systemic positive youth development models that enable resilience and growth. Emphasis is placed on promoting adaptive responses to PRD within school, family, and community systems in which the young person is embedded [68]. General facilitators of positive youth development may include, for example, teachers who are advocates and mentors, parenting programs that strengthen parent–child relationships, and community resources designed to promote equity of access to services and support [69].

There are also examples of evidence-based group programs focused on building coping and resilience. The Identity Project, validated through randomized controlled trials, has been shown to develop identity consolidation and improve psychological well-being of minority youths, through in-school workshops, team-building, and interpersonal interaction in a multi-week training format [70]. The Racial Encounter Coping Appraisal and Socialization Theory (RECAST) program focuses on the cognitive reappraisal and resolution of the discriminatory experience, coupled with family socialization and resilience support [71]. These and similar models have increasingly been applied to prevention and intervention with other vulnerable and marginalized youth, including immigrant, refugee, undocumented rural White youth, and the LGBTQ+ population [72]. Because such models are tailored to the specific needs of vulnerable individuals who have experienced discrimination and marginalization, they may be seen as more acceptable by young people and, therefore, serve as alternative or complementary approaches in relation to more traditional mental health interventions (e.g., crisis intervention and individual psychotherapy).

In Kentucky, efforts to address PRD and mental health among minority youths have been minimal, in part due to a changing political and educational landscape that has created obstacles to the implementation of such efforts in public school settings. Currently, to our knowledge, there are no state-funded initiatives that specifically address PRD among adolescents and only one recently funded program, the Black Youth Suicide Prevention Initiative, focused on the mental health of minority youths throughout the state. Kentucky’s prevention efforts have historically favored a universal rather than targeted prevention approach. However, the current study supports the need for a focused approach that integrates evidence-based programs addressing PRD and its relationship to adolescent mental health. For instance, initiatives aimed at fostering coping and resilience among minority youths and diminishing instances of PRD within schools and communities, similar to The Identity Project or RECAST. Immediate steps forward could include utilizing the results of this study to inform the Black Youth Suicide Prevention Initiative efforts and build public health capacity for reducing PRD and improving mental health outcomes within Kentucky’s youth population.

### 4.3. Limitations

The present findings should be interpreted in light of several limitations. Due to the cross-sectional nature of this study, causality between PRD and negative mental health cannot be inferred. The data are also susceptible to self-report bias, particularly regarding PRD, which may have been interpreted differently across racial and ethnic groups. Furthermore, the lack of data from the Jefferson County Public Schools, the state’s largest urban area, resulted in a reduced representation of Black students compared to statewide enrollment patterns. Consequently, this absence may have obscured the understanding of prevalence and correlates of PRD among Black students, limiting the generalizability of the findings to this population. Additionally, the larger proportions of sexual/gender minority and students attending schools in rural counties reporting SPD and suicidal behavior underscore the need for larger-scale studies, to comprehensively evaluate the intersectionality between race–ethnicity, sexual orientation, gender identity, and geography.

Finally, findings within this study portray outcomes for a demographic sample that diverges considerably from national studies, which are often drawn from more urban and suburban communities. For example, when contrasted with national school enrollment data for 2021, as reported by the National Center for Education Statistics, the Kentucky sample had a much higher proportion of White students (71.2% vs. 45.3% nationally) and a substantially lower representation from Black (6.3% vs. 15.0% nationally), Hispanic (8.8% vs. 29.1% nationally), and Asian (1.5% vs. 4.7% nationally) students [73]. While a strong association between PRD and negative mental health outcomes was confirmed, consistent with prior research, the pattern of prevalence rates and the strength of association diverged from other findings, likely due, in part, to demographics and the unique social and economic characteristics of Kentucky. It seems probable that similar variation would occur in other states and regions of the US, arguing for the need to gather and unpack surveillance data at regional and local levels when planning and implementing public health interventions focused on PRD and mental health.

### 4.4. Strengths

The current study has several strengths. The large sample size and resulting statistical power allowed for subgroup analyses, providing more in-depth insights into the association between PRD and mental health among a diverse group of adolescents. In particular, the availability of 6th-grade data provided the opportunity to explore the exposure and outcomes of interest in early adolescence, contributing to a better understanding of PRD from a developmental or life course perspective. Additionally, the predominantly rural nature of the study population provides a distinctive perspective often overlooked in large national samples, emphasizing the importance of understanding the influence of place on PRD and mental health outcomes among youth. Finally, the validation of the K6 scale to measure SPD within this population, as well as the comparability of suicide measures to other large and well-researched surveillance studies, enhances confidence in the findings and facilitates cross-study comparisons. Collectively, these strengths not only contribute to advancing theoretical understanding, but also hold significant implications for the development of targeted interventions aimed at mitigating the adverse impact of PRD on adolescent mental health and well-being.

## 5. Conclusions

The present study provides evidence of a direct relationship between PRD and the mental health and well-being of adolescents, with this association being most pronounced during the early rather than later stages of adolescence. These findings underscore the growing urgency for schools and communities to allocate resources towards evidence-based programs and interventions aimed at addressing PRD, ideally commencing in or before middle school. By prioritizing the development and implementation of culturally tailored interventions, there exists a significant opportunity to enhance youth well-being, while simultaneously fostering a more inclusive and supportive environment for all young people, irrespective of their race or ethnicity.

Furthermore, the imperative to tackle PRD among adolescents transcends individual well-being; it is indispensable for nurturing long-term social cohesion and stability. Discrimination not only inflicts harm upon its immediate targets, but also perpetuates cycles of inequality and societal division. By taking proactive measures to address PRD early in adolescents’ lives, we can contribute to nurturing a generation that is more resilient, empathetic, and empowered to confront injustice in all its forms. By equipping young people with the tools and support necessary to navigate and challenge discrimination, we can work towards building a more equitable and inclusive future for all.

## Figures and Tables

**Figure 1 healthcare-12-01011-f001:**
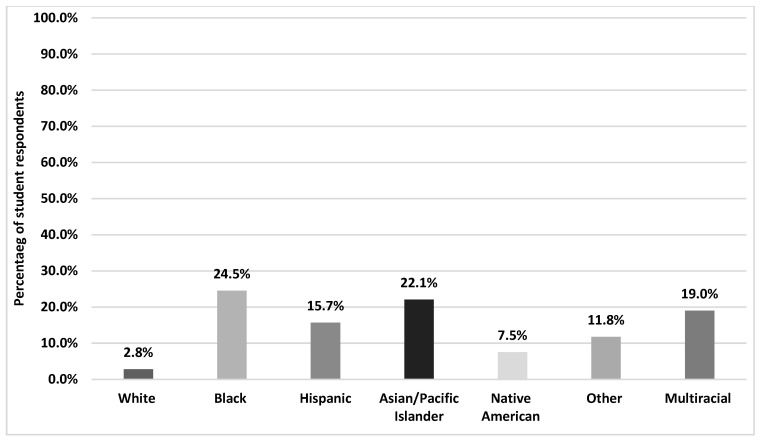
Prevalence of perceived racial discrimination by race and ethnicity.

**Figure 2 healthcare-12-01011-f002:**
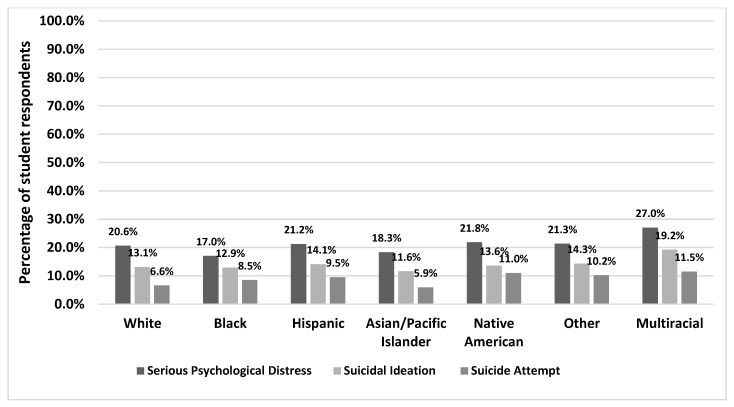
Prevalence of SPD and suicidality among racial and ethnic minorities.

**Figure 3 healthcare-12-01011-f003:**
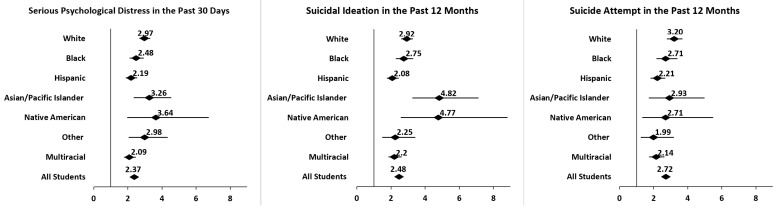
Adjusted odds of SPD and suicidality among racial and ethnic minorities reporting perceived racial discrimination.

**Table 1 healthcare-12-01011-t001:** Student characteristics.

		Overall		SPD ^c^		Suicide Ideation		Suicide Attempt	
Characteristics		*n*	(%)	*n*	(%)	*n*	(%)	*n*	(%)
Overall		93,812	(100)	19,310	(20.6)	12,524	(13.4)	6924	(7.4)
School Grade									
	6	25,552	(27.2)	3823	(14.8)	2443	(9.6)	1669	(6.5)
	8	26,712	(28.5)	5394	(20.2)	3736	(14.0)	2207	(8.3)
	10	23,632	(25.2)	5715	(24.2)	3694	(15.6)	1870	(7.9)
	12	17,916	(19.1)	4378	(24.4)	2651	(14.8)	1178	(6.6)
Gender Identity ^a^									
	Female	42,065	(44.8)	10,583	(25.2)	6,468	(15.4)	3592	(8.5)
	Male	46,052	(49.1)	5794	(12.6)	3,804	(8.3)	2068	(4.5)
	Questioning/unsure	1524	(1.6)	889	(58.3)	684	(44.9)	353	(23.2)
	Not listed	2103	(2.2)	1310	(62.3)	1047	(49.8)	580	(27.6)
	Prefer not to say	1756	(1.9)	684	(39.0)	486	(27.7)	304	(17.3)
Sexual Orientation ^a^									
	Heterosexual	62,888	(67.0)	9536	(15.2)	5600	(8.9)	2898	(4.6)
	Gay or lesbian	5444	(5.8)	2839	(52.2)	2247	(41.3)	1275	(23.4)
	Questioning/unsure	4343	(4.6)	1465	(33.7)	1031	(23.7)	530	(12.2)
	Not listed	6596	(7.0)	2964	(44.9)	2273	(34.5)	1196	(18.1)
	Prefer not to say	6547	(7.0)	1305	(19.9)	676	(10.3)	458	(7.0)
Race–Ethnicity ^b^									
	NH White (White)	66,829	(71.2)	13,763	(20.6)	8718	(13.1)	4430	(6.6)
	NH Black (Black)	5920	(6.3)	1007	(17.0)	761	(12.9)	504	(8.5)
	Hispanic	8265	(8.8)	1753	(21.2)	1165	(14.1)	788	(9.5)
	NH Asian/NH/PI (Asian/Pacific Islander)	1375	(1.5)	251	(18.3)	162	(11.6)	81	(5.9)
	NH AI/AN (Native American)	918	(1.0)	200	(21.8)	125	(13.6)	101	(11.0)
	NH Other (Other)	1743	(1.9)	372	(21.3)	249	(14.3)	177	(10.2)
	NH Multiracial (Multiracial)	5619	(6.0)	1518	(27.0)	1076	(19.2)	647	(11.5)
Rural									
	Yes	52,087	(55.5)	11,082	(21.3)	7144	(13.7)	4061	(7.8)
	No	41,725	(44.5)	8228	(19.7)	5380	(12.9)	2863	(6.9)
Perceived Racial Discrimination ^a^									
	Yes	5575	(5.9)	2140	(38.4)	1565	(28.1)	951	(17.1)
	No/Not sure	77,806	(82.9)	15,262	(19.6)	9770	(12.6)	5077	(6.5)

^a^ Overall characteristics may not add up to 100%, due to missing data. ^b^ NH = Non-Hispanic; NH/PI = Native Hawaiian or Pacific Islander; AI/AN = American Indian or Alaska Native. ^c^ SPD = Serious psychological distress.

## Data Availability

The 2021 Kentucky Incentives for Prevention dataset presented in this article is not readily available because the data are the property of the Kentucky Cabinet for Health and Family Services. Requests to access this dataset should be directed to Lisa Crabtree at lisa@reacheval.com. The Kentucky Department of Education dataset presented in the Appendix A is openly available at https://www.kyschoolreportcard.com/datasets?year=2021 (accessed on 2 February 2024).

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
