# Peer review of "Perceived Racial Discrimination, Psychological Distress, and Suicidal Behavior in Adolescence: Secondary Analysis of Cross-Sectional Data from a Statewide Youth Survey"

_healthcare, 2024, doi:10.3390/healthcare12101011_

Round 1

Reviewer 1 Report

Comments and Suggestions for Authors

In the attachment.

Reviewer 2 Report

Comments and Suggestions for Authors

The document is impressively organized and presented with great clarity.

The study design is thorough and includes comprehensive findings.

The strengths and limitations of the study are articulated with clarity and precision.

I found reading this document to be a gratifying experience.

More comments:

Introduction

consistent pattern was found in which PRD among minority adolescent subpopulations

was associated with higher levels of depression and internalizing symptoms, psychological distress, lower achievement and engagement, externalizing behaviors, risky behavior, substance abuse, and association with deviant peers.

Can you please expand on the internal symptoms?It would be worth mentioning here

A review of 138 empirical population studies of selfreported racism and various health/mental health outcomes, after adjusting for confounding variables, found the strongest and most consistent evidence emerged for associations between PRD and psychological distress and depressive symptom.

Could you clarify which confounding variables were taken into account?

Limitations

The authors did not discuss the hormonal changes that occur during youth. These changes are significant, especially in females, as they can contribute to mood swings. Boys may experience increased energy levels and mild aggression.

 It is worth noting that adolescents have access to substances, and marijuana is often a gateway drug. Were the students tested for drugs during the study?

Family history of suicide, temperament, personality, and epigenetics all play significant roles in this context. Some of the adolescents may have already received diagnoses for mental health disorders, while others may have recently migrated from Asian or African countries.

Conclusion

The authors can improve the conclusion by providing more details about the interventions.

For example, they could suggest referring individuals to a therapist or psychiatrist for additional assistance. They could also mention screening tools like the PHQ9 and recommend conducting drug screenings when necessary.

 Additionally, educating individuals about changes in mood and aggression during puberty could be helpful.

When migrants move to a new country, they must adapt to a new culture, societal norms, and ways of life. This process can be challenging and often takes several years to accomplish. During this time, they may experience cultural shock and face difficulties understanding the new environment. It is crucial to provide education and support to help them in this process.

 This can include language classes, cultural orientation programs, and resources to help them integrate into their new communities.
